# IMPROVING PRIVATE TRAINING VIA IN-DISTRIBUTION PUBLIC DATA SYNTHESIS AND GENERALIZATION

## ABSTRACT

To alleviate the utility degradation of deep learning classification with differential privacy (DP), employing extra public data or pre-trained models has been widely explored. Recently, the use of in-distribution public data has been investigated, where a tiny subset of data owners share their data publicly. In this paper, to mitigate memorization and overfitting by the limited-sized in-distribution public data, we leverage recent diffusion models and employ various augmentation techniques for improving diversity. We then explore the optimization to discover flat minima to public data and suggest weight multiplicity to enhance the generalization of the private training. While assuming 4% of training data as public, our method brings significant performance gain even without using pre-trained models, i.e., achieving 85.78% on CIFAR-10 with a privacy budget of $\varepsilon = 2$ and $\delta = 10^{-5}$.

## 1 INTRODUCTION

Differential privacy (DP) (Dwork, 2006; Dwork et al., 2014) establishes a mathematical framework to ensure the privacy of training data. In deep learning, differentially private stochastic gradient descent (DP-SGD) (Abadi et al., 2016) has become the de facto standard method. However, private deep learning with DP-SGD inevitably degrades performance compared to standard (non-DP) training (Bagdasaryan et al., 2019). As a practical solution, leveraging *public pre-trained models* or *public data* has been explored to enhance utility (Yu et al., 2021a;b; 2022; Li et al., 2022b; De et al., 2022; Tramer & Boneh, 2021). As using public data raises no privacy concerns, fine-tuning pre-trained models on private data can make use of learned features for free. However, since public data might be *out-of-distribution* in terms of private data, the utilization is restricted to when the public data have a similar distribution with private datasets (Tramer & Boneh, 2021).

Recently, researchers have investigated the use of *in-distribution public data*, indicating that a small portion of in-distribution data is made public (Li et al., 2022a). For example, some data owners may choose to share their data publicly in exchange for economic incentives. This setup allows us to leverage public data with a distribution similar to private data. Past studies have shown that utilizing side information from the in-distribution public during optimization can enhance the performance of DP-SGD (Nasr et al., 2023; Amid et al., 2022; Li et al., 2022a; Asi et al., 2021). However, repeated use of limited-sized public data may induce memorization and overfitting (Nasr et al., 2023).

We hypothesize that the performance of private learning can be further enhanced by leveraging in-distribution public data from two factors: first, the recent data synthesis model can enrich the side information of public data; second, the better optimization techniques concerning the geometric properties of loss functions can relieve overfitting and discover well-generalizing minima. We aim to bring these techniques into private learning, specifically dealing with the problem of the limited size of in-distribution public data, thereby achieving a new state-of-the-art classification performance.

The rest of the paper is organized as follows: Section 2 presents the background and related works, and Section 3 introduces the general scenarios using public data. In Section 4, we examine the risks of the generative model with limited data and employ current diffusion models (Karras et al., 2022; Kim et al., 2023a) and augmentation techniques as a solution. Section 5 utilizes geometric-based optimization (Foret et al., 2020) to alleviate public data overfitting. Furthermore, we introduce a method called *weight multiplicity* to enhance loss function smoothness (Park et al., 2023; Wang et al., 2021). In Section 6, we present the experimental results assuming 4% of public data, indicating that our results outperform the existing state-of-the-art methods, i.e., increasing the accuracy from

Table 1: Ablation study on the impact of various techniques, including synthesis, augmentation, and optimization, to enhance classification performance using in-sample public data from CIFAR-10 under $(2, 10^{-5})$-DP. Refer to the relevant sections for additional details.

| Setup | Training Settings | Section | Test Acc |
|---|---|---|---|
| **Baselines** | | | |
| Cold | Existing Baseline (WRN16-4, De et al. (2022)) | (3) | 64.02% |
| Warm | Warm-up on public data (warm) | (3) | 68.09% |
| WarmSyn | Warm-up on DDPM synthesis (Nasr et al., 2023) | (3) | 72.0% |
| Extended | DOPE-SGD (Nasr et al., 2023) | (3) | 75.1% |
| **In-sample Public Data Synthesis & Augmentation** | | | |
| WarmSyn | Warm-up on better synthetic data using EDM | (4.2) | 75.13 % |
| WarmSyn | EDM synthesis + Intra-class diversity (DG) | (4.3) | 77.66 % |
| WarmSyn | EDM synthesis + Augmentation (common + cutout) | (4.3) | 84.88% |
| **Generalization & Optimization** | | | |
| WarmSyn | + Sharpness-aware training in warm-up | (5.1) | 85.28% |
| Extended | + Weight multiplicity | (5.2) | 85.78% |

75.1% (Nasr et al., 2023) to 85.78% on CIFAR-10, with a privacy budget of $\varepsilon = 2$ and $\delta = 10^{-5}$. We first summarize our various approaches and their performance improvements in Table 1.

## 2 BACKGROUND AND RELATED WORK

### 2.1 DIFFERENTIALLY PRIVATE DEEP LEARNING

Differential privacy (DP) (Dwork et al., 2014) can guarantee the privacy of training data as follows:

**Definition 2.1** *(Differential privacy) For two adjacent inputs $d, d' \in \mathcal{D}$, a randomized mechanism $\mathcal{M} : \mathcal{D} \to \mathcal{R}$ satisfies $(\varepsilon, \delta)$-differential privacy for any set of possible outputs $\mathcal{S} \subseteq \mathcal{R}$ if*

$$Pr[\mathcal{M}(d) \in \mathcal{S}] \le e^{\varepsilon} Pr[\mathcal{M}(d') \in \mathcal{S}] + \delta. \tag{1}$$

The privacy budget $\varepsilon \ge 0$ controls the level of privacy guarantee with the broken probability $\delta \ge 0$. DP-SGD (Abadi et al., 2016) enables the private weight update by two steps: averaging the clipped per-sample gradient $\nabla \ell_i(\boldsymbol{w}) \coloneqq \nabla \ell(\boldsymbol{w}; \boldsymbol{x}_i)$ for weight $\boldsymbol{w}$ with respect to each individual private data sample $\boldsymbol{x}_i \in \boldsymbol{X}_t^{pr}$ and adds Gaussian noise to the averaged gradient, which is formulated as follows:

$$\tilde{\boldsymbol{g}}_t^{pr} = \boldsymbol{g}_t^{pr} + \mathcal{N}(\boldsymbol{0}, C^2\sigma^2\mathbf{I}) = \frac{1}{|\boldsymbol{X}_t^{pr}|} \sum_{\boldsymbol{x}_i \in \boldsymbol{X}_t^{pr}} \texttt{clip}\left(\nabla \ell_i(\boldsymbol{w}_t), C\right) + \mathcal{N}(\boldsymbol{0}, C^2\sigma^2\mathbf{I}), \tag{2}$$

where $\texttt{clip}(\boldsymbol{u}, C)$ projects $\boldsymbol{u}$ to the $L_2$-ball of radius $C$ and vector norm $\|\cdot\|$ means the $L_2$-norm $\|\cdot\|_2$. Then, we can update the weight by $\boldsymbol{w}_{t+1} = \boldsymbol{w}_t - \eta\tilde{\boldsymbol{g}}_t^{pr}$, where $\eta$ is the learning rate. The noise level $\sigma$ is determined by the privacy budget $(\varepsilon, \delta)$, the number of training steps, and the sampling probability (refer to Appendix A.1 for details). Additionally, we denote the gradient of public data as $\boldsymbol{g}_t^{pub} \coloneqq \frac{1}{|\boldsymbol{X}_t^{pub}|} \sum_{\boldsymbol{x}_i \in \boldsymbol{X}_t^{pub}} \nabla \ell_i(\boldsymbol{w}_t)$ without clipping and noise addition.

To mitigate the accuracy drop in DP-SGD, various studies explored DP-friendly properties, including architecture (Tramer & Boneh, 2021; Cheng et al., 2022) and loss/activation functions (Shamsabadi & Papernot, 2023; Papernot et al., 2021). As private learning often exhibits training instability (Bu et al., 2023), uncovering smooth or flat minima is effective for generalization (Park et al., 2023; Wang et al., 2021). Notably, De et al. (2022) achieved superior performance using the relatively large WideResNet (WRN) model, employing techniques such as *augmentation multiplicity* to minimize the averaged loss of various augmentations $\mathcal{L}_i(\boldsymbol{w}_t) = \mathbb{E}_k[\ell(\boldsymbol{w}_t, \texttt{aug}^k(\boldsymbol{x}_i))]$, weight standardization (Qiao et al., 2019), and Exponential Moving Average (EMA). Note that the recently proposed adaptive clipping (Andrew et al., 2021; Bu et al., 2023) could improve our results.

### 2.2 PRIVATE LEARNING WITH PUBLIC INFORMATION

Public pre-trained models offer the advantage of utilizing large models trained on extensive amounts of data (De et al., 2022). Thus, numerous researchers have shown their effectiveness in natural

language processing (NLP) (Yu et al., 2021a;b; 2022; Li et al., 2022b) and computer vision (De et al., 2022; Bu et al., 2022). The transfer learning using public data of similar distribution, such as CIFAR-100 and CIFAR-10, was also investigated (Tramer & Boneh, 2021; Sun & Lyu, 2021).

With in-distribution public data, previous studies (Nasr et al., 2023; Amid et al., 2022; Li et al., 2022a; Asi et al., 2021) have focused on reducing errors of DP-SGD with side information. These methods, categorized as *extended* in Section 3, utilize both public and private gradients during weight updates, defined as $\boldsymbol{w}_{t+1} = \boldsymbol{w}_t - \eta f_g(\boldsymbol{w}_t; \boldsymbol{X}_t^{pub}, \boldsymbol{X}_t^{pr})$. For example, normalizing private gradient $\boldsymbol{g}_t^{pr}$ with the accumulated public gradient $\boldsymbol{g}_t^{pub}$ (Li et al., 2022a) or linear combination of private gradient $\tilde{\boldsymbol{g}}_t^{pr}$ and public gradient $\boldsymbol{g}_t^{pub}$ (Amid et al., 2022) are proposed. Recently proposed DOPE-SGD (Nasr et al., 2023) first updates towards the public gradient $\boldsymbol{g}_t^{pub}$ and takes a private step towards $\boldsymbol{g}_t^{pub} - \boldsymbol{g}_t^{pr}$ to minimize the effects of clipping and noise addition as follows:

$$\boldsymbol{w}_{t+1} = \boldsymbol{w}_t - \eta(\boldsymbol{g}_t^{pub} + \texttt{clip}(\boldsymbol{g}_t^{pub} - \boldsymbol{g}_t^{pr}, C) + \mathcal{N}(\boldsymbol{0}, C^2\sigma^2\mathbf{I})). \tag{3}$$

From now on, we denote in-distribution public data as public data unless otherwise specified.

## 2.3 DIFFUSION SYNTHESIS FOR CLASSIFICATION AND PRIVACY

The integration of generative models in classification tasks has been widely explored to enhance generalization performance without extra data samples (He et al., 2023; Azizi et al., 2023; Gao et al., 2023). These models aim to generate diverse images within the data manifold, providing a better approximation of the decision boundary. The recent success of diffusion models has further advanced these approaches, providing both high quality and diversity in generated samples (Ho et al., 2020; Karras et al., 2022). However, most prior studies have concentrated on large data samples, such as ImageNet (Azizi et al., 2023) or the whole dataset of CIFAR-10 and CIFAR-100 (Wang et al., 2023). In the context of limited data availability, He et al. (2023) emphasized the significance of the diversity and data amount for generation to improve the classification performance.

To measure the quality of generated images, various measures are proposed: Inception Score (IS) (Salimans et al., 2016) and Precision to measure fidelity, Recall to measure diversity (Kynkäänniemi et al., 2019), Fréchet Inception Distance (FID) (Heusel et al., 2017) to measure the distributional quality for mean and variance. However, as the synthesis measures can be not aligned with classification performance, Ravuri & Vinyals (2019) suggested the classification accuracy score (CAS) which measures the classification performance on a test set using a model trained on synthetic data.

In terms of the data synthesis for privacy, only Nasr et al. (2023) explored the use of the denoising diffusion probabilistic model (DDPM) (Ho et al., 2020) as an augmentation technique for in-distribution data, which closely aligns with our settings. Note that differentially private data synthesis with diffusion models are investigated (Dockhorn et al., 2022; Lyu et al., 2023), presenting huge opportunities for future work. For example, Ghalebikesabi et al. (2023) demonstrated that private fine-tuning on pre-trained diffusion models can improve both generation and classification results.

## 3 PROBLEM SETUP

**Setup**  We follow the most common scenarios leveraging public data (Nasr et al., 2023). In this paper, we specifically focus on the **WarmSyn** and **Extended** scenarios among the following settings:

- **Cold**: Conduct DP training (e.g., DP-SGD) on the private dataset without using public data.
- **Warm**: Train models through (i) non-DP *warm-up training* phase on the public dataset (e.g., SGD) and (ii) *private training* phase on the private dataset with DP methods.
- **WarmSyn**: During the warm-up phase, amplify the public information by using synthesis or augmentation methods. Then, train models in the same manner as in the Warm settings.
- **Extended**: After the warm-up training of WarmSyn, further utilize the gradient information from both the private dataset and the non-private dataset in the private training phase.

In Sections 4 and 5, we primarily focus on the CIFAR-10 dataset, utilizing WRN-16-4[1] with the techniques proposed in (De et al., 2022). Within the training set, we randomly select 2,000 instances

---

[1]As noted in (Nasr et al., 2023), training WRN-40-4 requires excessive GPU considering its performance.

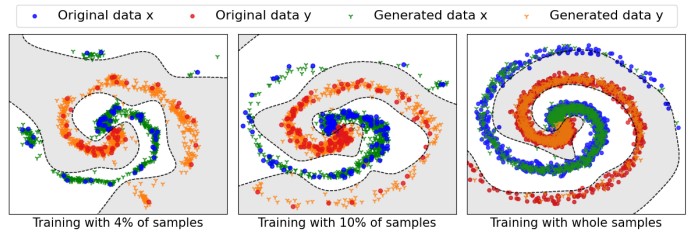
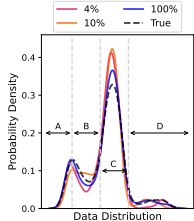

(a) Illustration of original(●)-generated(⊽) data and decision boundaries.  (b) Probability density.

Figure 1: Diffusion toy experiment on spiral data by varying the portion of training samples.

(4%) for the public samples, as suggested in (Nasr et al., 2023). These samples are uniformly drawn from each class. Additional experimental details are provided in Section 6.1 and Appendix C.

# 4 DATA SYNTHESIS FOR IN-DISTRIBUTION PUBLIC DATA

Given the restricted number of public data, we first analyze how to amplify the side information in the public data. To achieve this, we employ a better diffusion model to approximate the underlying true distribution based on the public data. Subsequently, we investigate the diversity of synthetic data to enhance their utility for classification tasks. All the measures for evaluating the generated data in this section are calculated with the entire training set as a reference.

## 4.1 RISK OF GENERATIVE MODELS WITH LIMITED DATA

We first describe that the generative models with a limited number of training samples can possess risks, i.e., over- or underestimating the data distribution. As a toy example, we depict a spiral dataset with higher probabilities for points closer to the origin, detailed in Appendix D.1. We train the diffusion models with 100, 250, and 2,500 training samples. Subsequently, we generate 1,000 synthetic data from each model and draw decision boundaries with SVM using generated data. As shown in Figure 1a, the generated data successfully captures the data manifold in dense regions, but they fail to find the manifold and only imitate the training data in sparse regions. As shown in the probability density functions along the x-axis in Figure 1b, the models trained with small samples possess higher probabilities than the true distribution in the dense region (C). Conversely, their generated data have lower probabilities than the true distribution in sparse regions (B and D). The lack of diversity in Figure 1a may lead to misclassification due to distorted decision boundaries.

## 4.2 BETTER DIFFUSION MODELS FOR PUBLIC DATA SYNTHESIS

In the recent work, Nasr et al. (2023) utilized DDPM as an augmentation for public data to improve classification performance. We now demonstrate that a more accurate approximation within the in-distribution data can enhance the model's ability to match the whole data distribution.

**Theorem 4.1** *(Distribution matching with in-distribution data) For a finite number of training data samples $S_{data} = \{x_1, \cdots, x_N\}$, split the data samples into $S_{pub} = \{x_1, \cdots, x_n\}$ and $S_{pr} = \{x_{n+1}, \cdots, x_N\}$, without loss of generality. Let $p_{data}, p_{pub},$ and $p_{pr}$ be the probability distribution of the corresponding dataset and $p_\theta$ be a data distribution generated by the trained model. Then,*

$$r_1 \log r_1 + r_2 \log r_2 \le D_{KL}(p_{data}\|p_\theta) - D_{KL}(p_{pub}\|p_\theta) \le r_1 \log r_1 + r_2 \log r_2 + r_2 \log \tau_{pr},$$

*where $\tau_{pr} = \{(N-n) \min_{x \in S_{pr}} p_\theta(x)\}^{-1} \ge 1$. $r_1 = \frac{n}{N}$ and $r_2 = \frac{N-n}{N}$ represent the ratios of public and private data, respectively. Therefore, when $r_1$ is small, $p_\theta$ that minimizes $D_{KL}(p_{pub}\|p_\theta)$ can approximate $p_{data}$ and $p_{pr}$. When $r_1 \to 1$, then $p_{pub} \to p_{data}$ and the equality holds.*

The detailed proof is presented in Appendix B. Thus, to better generate synthetic images by minimizing $D_{KL}(p_{pub}\|p_\theta)$, we adopt the elucidating diffusion model (EDM) (Karras et al., 2022) that achieves the lowest FID scores. We train the EDM model on 2,000 public data samples without using external datasets and employ class-conditional sampling to match the original distribution. The

Table 2: Quality comparison of synthetic data trained with 4% of public data on CIFAR-10.

| Sampling | Fidelity | | Diversity | Quality | CAS (↑) | Test Acc |
| | IS (↓) | Precision (↑) | Recall (↑) | FID (↓) | (%) | (%) |
|---|---|---|---|---|---|---|
| EDM ($w_d$=0) | **11.008** | **0.964** | 0.157 | 7.799 | 62.82 | 75.13 |
| EDM + DG ($w_d$=3) | 10.815 | **0.964** | 0.153 | **7.786** | 62.47 | 75.31 |
| EDM + DG ($w_d$=10) | 10.796 | 0.946 | 0.170 | 8.274 | 64.61 | 75.98 |
| EDM + DG ($w_d$=20) | 10.157 | 0.873 | 0.191 | 11.497 | **67.61** | **77.66** |
| EDM + DG ($w_d$=30) | 9.113 | 0.785 | **0.211** | 19.748 | 66.53 | 77.22 |

calculated measures with EDM samples are shown in Table 2. The FID score of EDM synthesis at 7.80 outperforms the reported FID score of 12.8 achieved with DDPM on 40K images (Nasr et al., 2023), resulting in improved classification performance as shown in Table 1. However, the generation quality using public data is notably worse than the FID of 1.79 achieved with the entire dataset, as reported in (Karras et al., 2022). Within the limited public data, the model struggles to capture diversity, resulting in a recall of 0.16, even though precision remains high at 0.96. Note that repeating each sample in the public data 25 times (thus 50,000 samples) results in a precision of 1.00, recall of 0.04, and FID of 13.64. The generated images and its memorization problems are illustrated in Appendix F. Thus, we need to enhance the diversity within the generated images.

### 4.3 INTRA-CLASS DIVERSITY MATTERS

**Diversity in generation**    To enhance the intra-class diversity during generation, we adopt the idea of discriminator guidance (DG) (Kim et al., 2023a). By introducing a discriminator to judge whether the sampling is from the true data or synthesis, DG can control the trade-off between fidelity and diversity of the generated images by adjusting the weight $w_d$ of the discriminator. Refer to Appendix D.2 for the details of DG. As shown in Table 2, a higher weight of guidance ensures greater intra-class diversity without any augmentation, but sacristies the fidelity. The best FID score is obtained with $w_d = 3$ while the best CAS is obtained at a bigger weight $w_d = 20$.

As observed in (He et al., 2023; Ravuri & Vinyals, 2019), diversity plays an important role in improving CAS. Interestingly, we need to larger the weight in the in-sample data than that of standard training, to achieve a similar gain of diversity. Figure 2 represents selected examples from two extreme cases, with $w_d = 0$ and $w_d = 30$ to visualize the difference. Despite the quality degradation of detailed features with $w_d = 30$, the CAS is higher due to the increased intra-class diversity in features. Thus, we need to enhance diversity while maintaining quality for better classification.

**Explicit diversity with data augment**    To explicitly enhance diversity in synthetic images during the warm-up training, we employ various data augmentation techniques designed for classification tasks. Wang et al. (2023) argued that utilizing appropriate augmentation in diffusion-based generated images can further improve the classification performance. Common augmentation (He et al., 2016) uses padding and random crop to the original size and horizontal flipping for the images. Cutmix (Yun et al., 2019) randomly replaces a part of the image with another and cutout (DeVries & Taylor, 2017) randomly pad images. AutoAugment (Cubuk et al., 2019) chooses the best combination of augmentations such as color, rotation, or cutout. We set a baseline with no augmentation for synthetic images of EDM and then apply the aforementioned augmentations. The results are summarized in Table 3, where adding cutout augmentation to common augmentation demonstrates the best performance in terms of CAS and test accuracy. Interestingly, combining DG and augmentation does not yield further improvement, as presented in Appendix E.1.

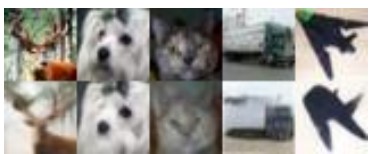

| Augmentation | CAS(%) | Test Acc (%) |
|---|---|---|
| EDM w/o augmentation | 64.02 | 75.32 |
| Common | 77.99 | 83.97 |
| Common + Cutout | **80.72** | **84.88** |
| Common + Cutmix | 65.73 | 82.20 |
| AutoAugment | 77.21 | 83.45 |

Figure 2: Selected samples from (top) EDM and (bottom) EDM + DG ($w_d = 30$).

Table 3: CAS (%) and test accuracy (%) with different augmentation methods during warm-up phase.

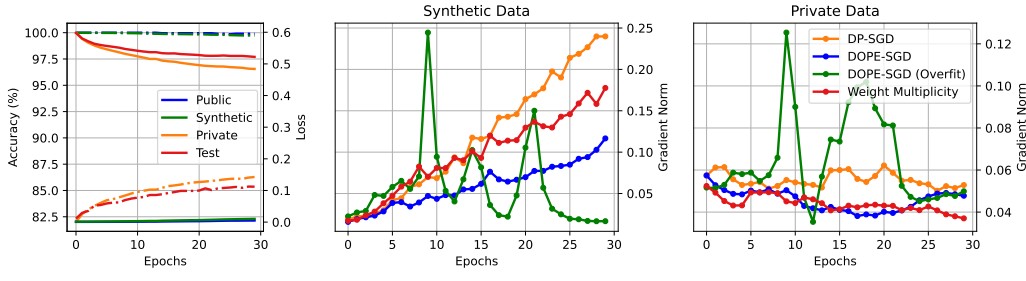

(a) Accuracy (−·) and loss (−).     (b) Gradient norm w.r.t (left) synthetic data and (right) public data.

Figure 3: Learning dynamics of private training with different datasets and optimization methods.

# 5 GENERALIZATION OF PRIVATE OPTIMIZATION

## 5.1 MITIGATING OVERFITTING TO PUBLIC DATA

We now investigate the optimization properties of private training after the warm-up phase of Section 4. In Figure 3a, we illustrate the accuracy and loss for each public, synthetic, private, and test dataset during private training. Notably, the model consistently achieves near-zero loss and 100% accuracy on synthetic data (also on public data) but struggles with private data. This implies that we need to focus on private data during optimization as the model has already learned the most of information from public data. Figure 3b illustrates the gradient norms of synthetic and public data using different optimization methods, with each gradient calculated based on the corresponding batch size. While the private gradient norm $\|g_t^{pr}\|$ (clipped but without noise) gradually decreases, the public gradient norm $\|g_t^{pub}\|$ consistently increases across all optimization methods. Interestingly, due to the large learning rate of private training and advanced warm-up models using EDM and augmentation, DOPE-SGD can encounter exploding gradients. This occurs when the model is overfitted and stuck into sharp minima on synthetic data and diverges after updating towards synthetic gradients without clipping in Equation (3). For further insights about the learning dynamics in terms of the loss function and private gradient norm without clipping, please refer to Appendix D.3.

Therefore, we focus on the geometric properties of the loss landscape in the warm-up phase, specifically *sharpness* and *smoothness* (Dinh et al., 2017; Keskar et al., 2017). To prevent overfitting to public information, we consider using sharpness-aware minimization (SAM) (Foret et al., 2020) on the warm-up training. Instead of minimizing $w_t$, SAM effectively identifies flat minima by minimizing the worst-case perturbation within a parameter space of radius $\rho$ as follows:

$$w_{t+1} = w_t - \eta\nabla\ell(w_t^p) = w_t - \eta\nabla\ell(w_t + \rho\nabla\ell(w_t)/\|\nabla\ell(w_t)\|). \tag{4}$$

The results presented in Table 4 show the effectiveness of SAM (4) in the warm-up phase, obtaining lower values in the top Hessian value $\lambda_{max}$, ratio of Hessian $\lambda_{max}/\lambda_5$, and trace of Hessian matrix $\text{Tr}(\nabla^2\ell(w))$. Specifically, SAM uncovers flat minima, not only with synthetic data but also with private data. Note that SGD even demonstrates a negative trace value on private data.

Table 4: Geometric measures of the models trained with SGD and SAM after warm-up phase.

| Optimizer | Synthetic Data | | | Private Data | | |
|---|---|---|---|---|---|---|
| | $\lambda_{max}$ | $\lambda_{max}/\lambda_5$ | $\text{Tr}(\nabla^2\ell(w))$ | $\lambda_{max}$ | $\lambda_{max}/\lambda_5$ | $\text{Tr}(\nabla^2\ell(w))$ |
| SGD | 1.38 | 10.62 | 71.98 | 112.82 | 1.58 | -2527.78 |
| SAM | **0.44** | **2.63** | **53.21** | **58.32** | **1.50** | 333.03 |

## 5.2 WEIGHT MULTIPLICITY

To push further, we pursue to improve generalization by seeking a flat and smooth loss landscape in the private training phase. To discover flat minima in private learning, minimizing the weights in the vicinity of the parameter space can be a practical solution (Wang et al., 2021; Park et al., 2023). Thus, we introduce an effective optimization concept for seeking flat and smooth minima,

which we call *weight multiplicity*, to minimize $\mathcal{L}_i(\boldsymbol{w}) = \mathbb{E}_{\boldsymbol{v}}[\ell(\boldsymbol{w} + \boldsymbol{v}; \boldsymbol{x}_i)]$ with the perturbation vector $\boldsymbol{v}$ sampled from the parameter space. This idea is analogous to the augmentation multiplicity (De et al., 2022) to minimize $\mathbb{E}_k[\ell(\boldsymbol{w}; \text{aug}^k(\boldsymbol{x}_i))]$ for each unique example, which is widely used in recent studies on DP-SGD (Nasr et al., 2023; Knolle et al., 2023; Ghalebikesabi et al., 2023) with $K \geq 16$ multiplicity. Therefore, we combine the two aforementioned methods and focus on minimizing $\mathbb{E}_k[\ell(\boldsymbol{w} + \boldsymbol{v}^k; \text{aug}^k(\boldsymbol{x}_i))]$ to simultaneously improve the generalization of both input-loss and weight-loss landscapes. Then, we can alter the private gradient of Equation (2) as follows:

$$\boldsymbol{g}_t^{pr} = \frac{1}{|\boldsymbol{X}_t^{pr}|} \sum_{\boldsymbol{x}_i \in \boldsymbol{X}_t^{pr}} \texttt{clip}\left(\frac{1}{K} \sum_{k=1}^{K} \nabla\ell\left(\boldsymbol{w}_t + \boldsymbol{v}_t^k; \text{aug}^k(\boldsymbol{x}_i)\right), C\right). \tag{5}$$

Motivated by (Gong et al., 2021), the following Theorem proves the effect of weight and augmentation multiplicity as a gradient norm regularization w.r.t the weight and input space, respectively.

**Theorem 5.1** *(Multiplicity as gradient norm regularizer) Let $\mathcal{L}_m(\boldsymbol{w}, \boldsymbol{v}^1 : \boldsymbol{v}^K; \boldsymbol{x}_i) = \frac{1}{K}\sum_{k=1}^{K} \ell(\boldsymbol{w} + \boldsymbol{v}^k; \text{aug}^k(\boldsymbol{x}_i))$ with $\|\boldsymbol{v}^k\| = \rho$. For $\ell(\boldsymbol{w} + \boldsymbol{v}^k; \boldsymbol{x}_i) \geq \ell(\boldsymbol{w}; \boldsymbol{x}_i)$ and $\ell(\boldsymbol{w}; \text{aug}^k(\boldsymbol{x}_i)) \geq \ell(\boldsymbol{w}; \boldsymbol{x}_i)$,*

$$\mathcal{L}_m(\boldsymbol{w}, \boldsymbol{v}^1 : \boldsymbol{v}^K; \boldsymbol{x}_i) = \ell(\boldsymbol{w}; \boldsymbol{x}_i) + \Phi(\boldsymbol{w}, \boldsymbol{v}^1 : \boldsymbol{v}^K; \boldsymbol{x}_i) + \Psi(\boldsymbol{w}; \boldsymbol{x}_i) + O(\rho^2) + O(h^2)$$

$$\text{with } c_1 \sum_{k=1}^{K} \|\nabla_{\boldsymbol{w}}\ell(\boldsymbol{w}; \text{aug}^k(\boldsymbol{x}_i))\| \leq \Phi(\boldsymbol{w}, \boldsymbol{v}^1 : \boldsymbol{v}^K; \boldsymbol{x}_i) \leq c_2 \sum_{k=1}^{K} \|\nabla_{\boldsymbol{w}}\ell(\boldsymbol{w}; \text{aug}^k(\boldsymbol{x}_i))\|$$

$$\text{and } c_3\|\nabla_{\boldsymbol{x}}\ell(\boldsymbol{w}; \boldsymbol{x}_i)\| \leq \quad \Psi(\boldsymbol{w}; \boldsymbol{x}_i) \quad \leq c_4\|\nabla_{\boldsymbol{x}}\ell(\boldsymbol{w}; \boldsymbol{x}_i)\|,$$

*with some constants $c_i \geq 0$ and $h = \sum_{k=1}^{K} \|\text{aug}^k(\boldsymbol{x}_i) - \boldsymbol{x}_i\|/K$. Therefore, minimizing $\mathbb{E}_{\boldsymbol{x}}[\mathcal{L}_m]$ has the effect of adding extra Lipschitz-like regularization in terms of weight and input gradient norms.*

The detailed proof is presented in Appendix B. However, the optimal direction of perturbation $\boldsymbol{v}$ is still debated, considering privacy leakage (Park et al., 2023) or uncertain optimization properties (Compagnoni et al., 2023; Kim et al., 2023b). Thus, to extend the use of public information, we focus on the fact that gradient descent primarily occurs within a low-dimensional top Hessian subspace (Sagun et al., 2018; Papyan, 2019; Lee et al., 2023) and its effectiveness in private training (Yu et al., 2021b; Ye & Shokri, 2022). As depicted in Figure 4, we visualize the concentration of individual gradient directions for both synthetic and private data when projected onto a 2D plane spanned by the mean vectors of two classes (dog-bird). The results in a low-dimensional subspace suggest that individual gradient directions of both datasets are similar among intra-class samples but differ between classes. Therefore, we suggest to calculate the perturbation $\boldsymbol{v}_t^k$ of the private sample $\boldsymbol{x}_i$ with the public batch $\boldsymbol{X}_t^{pub}$, similar to Equation (4) as follows:

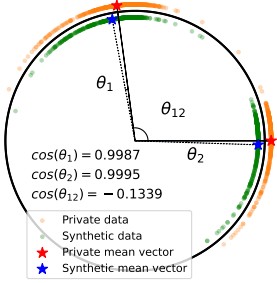

Figure 4: Individual gradient concentration of private and synthetic data on a 2D plane spanned by dog-bird classes.

$$\mathbb{E}_k[\ell(\boldsymbol{w}_t + \boldsymbol{v}_t^k; \text{aug}^k(\boldsymbol{x}_i))] = \frac{1}{K} \sum_{k=1}^{K} \ell\left(\boldsymbol{w}_t + \rho\frac{\nabla\ell(\boldsymbol{w}_t; \boldsymbol{X}_t^{pub_k})}{\|\nabla\ell(\boldsymbol{w}_t; \boldsymbol{X}_t^{pub_k}))\|}; \text{aug}^k(\boldsymbol{x}_i)\right). \tag{6}$$

Please refer to Appendix A.2 for the detailed algorithm. Due to the high similarities of individual gradients in Figure 4, Equation (6) can act as a regularization of the gradient norm to weight space, as proved in Theorem 5.1. Note that Equation (6) requires the same privacy budget as DP-SGD since calculating each ascent direction with a public gradient $\nabla\ell_i(\boldsymbol{w}_t; \boldsymbol{X}_t^{pub})$ does not reveal private data and averaging $K$ gradients for individual sample do not leakage privacy (De et al., 2022).

# 6 RESULTS

## 6.1 EXPERIMENTAL SETUP

We assess the effectiveness of our proposed methods using public data in two datasets: CIFAR-10 and CIFAR-100. For the public dataset, we randomly sample 4% of the training data (2,000 samples) uniformly drawn from each class, while the remaining data are used as private samples. We

Table 5: Test accuracy (%) of private classification on CIFAR-10 on privacy budget of $\varepsilon \in \{1, 2, 3, 4, 6\}$. The public and synthesis columns indicate the Warm and WarmSyn settings, respectively. Our method employs all the techniques in Table 1. We highlight the best accuracy in **bold**.

| Datasets | Architecture | Public | Synthesis | Method | $\varepsilon = 1$ | $\varepsilon = 2$ | $\varepsilon = 3$ | $\varepsilon = 4$ | $\varepsilon = 6$ |
|---|---|---|---|---|---|---|---|---|---|
| | CNN-Tanh (0.55M) | ✗ | ✗ | Papernot et al. (2021) | 45.8 | 58.3 | 63.5 | - | - |
| | ScatterNet (0.16M) | ✗ | ✗ | Tramer & Boneh (2021) | 60.3 | 67.2 | 69.3 | - | - |
| | DPNAS (0.53M) | ✗ | ✗ | Park et al. (2023) | 60.1 | 67.2 | 69.9 | - | - |
| | WRN-16-4 (2.74M) | ✗ | ✗ | De et al. (2022) | 56.8 | 64.9 | 69.2 | 71.9 | 77.0 |
| CIFAR | WRN-40-4 (8.94M) | ✗ | ✗ | De et al. (2022) | 56.4 | 65.9 | 70.7 | 73.5 | 78.8 |
| -10 | | ✓ | ✗ | Amid et al. (2022)[†] | - | 68.7 | - | 73.1 | 77.2 |
| | | ✓ | ✗ | Li et al. (2022a)[†] | - | 68.7 | - | 73.5 | 77.9 |
| | WRN-16-4 (2.74M) | ✓ | DDPM | Amid et al. (2022)[†] | - | 70.5 | - | 74.5 | 78.2 |
| | | ✓ | DDPM | Li et al. (2022a)[†] | - | 69.1 | - | 74.1 | 78.1 |
| | | ✓ | DDPM | Nasr et al. (2023)[†] | - | 75.1 | - | 77.9 | 80.0 |
| | | ✓ | EDM | **Ours** | **84.32** | **85.78** | **86.00** | **86.59** | **87.23** |

[†]We note the results reported in (Nasr et al., 2023) to set the architecture same. All other baseline results are adopted from the original paper.

Table 6: Test accuracy (%) of private classification on CIFAR-100 on the privacy budget of $\varepsilon \in \{1, 2, 6, 10\}$. The public and synthesis columns indicate Warm and WarmSyn settings, respectively. We employ the techniques in Table 1 sequentially, i.e., synthesis, augmentation, and optimization.

| Datasets | Architecture | Public | Synthesis | Methods | $\varepsilon = 1$ | $\varepsilon = 2$ | $\varepsilon = 6$ | $\varepsilon = 10$ |
|---|---|---|---|---|---|---|---|---|
| | Resnet-9 | ✗ | ✗ | Knolle et al. (2023)[†] | 18.1 | 24.9 | - | 40.8 |
| | Resnet-9 (6.62M) | ✗ | ✗ | Cold[†] | 8.35 | 14.42 | 29.89 | 35.11 |
| CIFAR | WRN-16-4 (2.74M) | ✗ | ✗ | Cold | 9.28 | 18.19 | 33.61 | 39.09 |
| -100 | | ✓ | ✗ | Warm | 20.84 | 25.15 | 33.47 | 38.89 |
| | WRN-16-4 (2.74M) | ✓ | EDM | WarmSyn | 26.27 | 31.55 | 35.61 | 40.89 |
| | | ✓ | EDM | +Augmentation | 41.17 | 44.51 | 50.50 | 54.25 |
| | | ✓ | EDM | +Optimization | **45.89** | **47.93** | **54.72** | **56.46** |

[†]Unfortunately, their DP-SGD results are not reproducible, even when using the same hyperparameters as in the original paper.

then train the EDM (Karras et al., 2022) models with the 2,000 public data samples and build 50K synthetic datasets with EDM sampling. For classification models, we primarily adopt WRN-16-4, following the methodology outlined in (De et al., 2022) and use pre-trained vision transformer models following (Bu et al., 2022). Our experiments are conducted using PyTorch libraries (Yousefpour et al., 2021) on eight NVIDIA GeForce RTX 3090 GPUs, partially performed on a cloud server with four NVIDIA A100 40GB GPUs. For more detailed settings, including learning rates, epochs, public batch sizes, and the radius of weight multiply $\rho$, refer to Appendix C. We will make the code public at `anonymized-url`, including the trained diffusion models and the sampled synthetic images.

## 6.2 CLASSIFICATION PERFORMANCE WITH PUBLIC DATA

**Effects of individual techniques** We first revisit Table 1, the ablation study of sequentially employing our techniques with privacy budget $(2, 10^{-5})$-DP on CIFAR-10. By using better EDM synthesis, we achieve the previous SOTA performance of 75.13%. Additionally, recognizing the significance of diversity in data generation, we employ common + cutout augmentation techniques for diversity in classification, resulting in a performance of 84.88%. To mitigate potential overfitting to side information, we make use of generalization techniques, such as SAM and the proposed weight multiplicity. All these efforts collectively lead to an accuracy of 85.78% under $(2, 10^{-5})$-DP.

**CIFAR-10 experiments** We report the performance comparison for our method with various previous approaches on a wide range of $\varepsilon \in \{1, 2, 3, 4, 6\}$ with $\delta = 10^{-5}$ in Table 5. Our approach, which incorporates EDM synthesis, augmentation, and optimization through SAM and weight multiplicity, exhibits superior classification performance when compared to existing methods including DDPM-based augmentation and previous extended optimizations. The accuracies of $\varepsilon = 0$ (after warm-up, without private data) and $\varepsilon = \infty$ (not private, with clipping) are 80.4% and 88.5%, respectively.

**CIFAR-100 experiments** We then investigate a more complex dataset of CIFAR-100, where the classification from scratch without using pre-trained models is not actively investigated. Similar to

Table 7: Test accuracy (%) of private classification using pre-trained models.

| Datasets | Privacy budget | | | $\varepsilon = 0.5$ | | | $\varepsilon = 2$ | | |
|---|---|---|---|---|---|---|---|---|---|
| | Architecture | Pre-trained | Cold | Warm | Ours | Cold | Warm | Ours |
| CIFAR -100 | CrossViT small 240 (26.3M) | ✓ | 61.70 | 73.34 | **77.52** | 70.99 | 77.19 | **80.66** |
| | CrossViT 18 240 (42.6M) | ✓ | 67.02 | 77.50 | **80.03** | 78.61 | 80.31 | **82.91** |
| | DeiT base patch16 224 (85.8M) | ✓ | 49.34 | **80.08** | 79.81 | 69.09 | 83.01 | **83.07** |
| | CrossViT base 240 (103.9M) | ✓ | 65.90 | 75.45 | **78.09** | 75.13 | 79.21 | **81.25** |

Table 8: Performance by varying the number of synthetic data.

Table 9: Performance and computational time of optimization methods. **Bold** for best and underline for runner-up results.

| Generated | Test Acc (%) | | |
|---|---|---|---|
| | $\varepsilon = 2$ | $\varepsilon = 4$ | $\varepsilon = 6$ |
| 5K | 77.48 | 77.98 | 78.00 |
| 20K | 83.29 | 84.08 | 84.22 |
| 40K | **85.14** | **85.61** | **86.03** |

| Privacy budget $\varepsilon$ | Optimization | | | |
|---|---|---|---|---|
| ($\delta = 10^{-5}$) | DP-SGD | Mirror GD | DOPE-SGD | Weight multiplicity |
| $\varepsilon = 2$ | 85.28 | 85.52 | 85.53 | **85.78** |
| $\varepsilon = 4$ | 86.55 | 86.31 | 86.69 | **86.71** |
| $\varepsilon = 6$ | 87.08 | 86.73 | 86.84 | **87.23** |
| Time (ms/image) | **12.70** | 12.80 | 12.86 | 13.06 |

CIFAR-10, we first train the EDM model with 2K images and generate 50K images. Given the 100 classes in CIFAR-100, only 20 public samples are available per class, significantly fewer than in CIFAR-10. The FID score on 50K images is 11.28 on EDM synthesis and 15.58 on replicating each public image 25 times. After the warm-up, the CAS of synthetic images is 38.04%, while the test accuracy of using 2K public samples without synthesis is only 16.79%. In Table 6, all the approaches in Table 1 including synthesis, augmentation, and optimization are sequentially adopted. As a result, we succeed to obtain 47.93% accuracy on $\varepsilon = 2$ without using pre-trained models.

**CIFAR-100 with pre-trained models**  To push further, we demonstrate the effectiveness of our procedures when combined with pre-trained models. We adopt the vision transformers model for fine-tuning without augmentation-weight multiplicity, aligning with the original implementation (Bu et al., 2022). The results in Table 7 indicate that using in-sample data can boost the classification performance. Within a wide range of models and privacy budgets, our methods outperform the warm settings even though private learning is sensitive to training settings.

## 6.3 SENSITIVITY ANALYSIS

**Amount of generated data**  To analyze the effect of the generated data size, we train models with 5K, 20K, and 40K synthetic samples in the warm-up phase until convergence for each model. The results in Table 8 indicate that data size remains a critical factor, even when applying the identical EDM model. Note that 40K sample size is equal in the experiments of (Nasr et al., 2023).

**Effects of extended methods**  In Table 9, we compare the effects of different optimization methods, which include DP-SGD, mirror GD (Amid et al., 2022), and DOPE-SGD (Nasr et al., 2023), while maintaining the same synthesis, augmentation, and sharpness-aware warm-up techniques. Interestingly, the performance improvements from existing extended methods are not substantial. In contrast, our proposed weight multiplicity approach demonstrates the potential to enhance performance, with a consistent decrease in private gradient norm, as illustrated in Figure 3b. The computational time in Table 9, indicates minimal overhead (less than 3%) when employing extended methods with a multiplicity of $K = 16$. Additional ablation studies can be found in Appendix E.

## 7 CONCLUSION

In this paper, we investigated the potential of using in-distribution public data in differentially private classification tasks. By using the current diffusion generative models and augmentation techniques, we demonstrated the importance of diversity. To relieve overfitting to the public data, we found well-generalizing minima by using the geometric properties. As a limitation, we leave the experiments on sensitive real-world datasets such as medical or face image datasets for future work. We believe that this work can contribute to leveraging in-distribution data for relieving utility-privacy trade-offs.

ETHICS STATEMENT

When using in-distribution public data, we should be aware of the undesirable privacy leakage of private data. Moreover, using pre-trained models may also introduce another privacy risk, when combined with other forms of sensitive data, such as language models or time series data. Therefore, we can consider the potential of our methods with additional privacy-preserving techniques, such as federated learning or homomorphic encryption.

REPRODUCIBILITY

We provide the code for reproduction in the supplementary material. Please refer to Appendix C for experimental settings. Due to the size limits, we will provide the trained diffusion models and sampled images after submission.

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

# A DIFFERENTIAL PRIVACY

## A.1 DP-SGD

The noise level $\sigma$ of DP-SGD (2) is determined by the total steps, sampling probability, and privacy budget $(\varepsilon, \delta)$ as follows:

**Proposition A.1** *(Abadi et al. (2016)). There exist constant $c_1$ and $c_2$ so that given total steps $T$ and sampling probability $q$, for any $\varepsilon < c_1 q^2 T$, DP-SGD (2) guarantee $(\varepsilon, \delta)$-DP, for any $\delta > 0$ if we choose*

$$\sigma \geq c_2 \frac{q\sqrt{T\log(1/\delta)}}{\varepsilon}. \tag{7}$$

## A.2 WEIGHT MULTIPLICITY

De et al. (2022) modified the gradient of DP-SGD (2) with $K$ multiple augmentations to calculate the individual gradient. Inspired by augmentation multiply, we further investigate the average of multiple weights space for smoothness and flatness in Equation (5). For selecting the direction $v$ for perturbation, we choose to use the public gradient as a line of the extended method. The detailed algorithm of Equation (6) is presented in Algorithm 1. As the weight-augmentation multiplicity only needs the computation for calculating $v_t^k$ for weight multiplicity, its computational burden is marginal with respect to augment multiplicity. Note that the direction of $v$ can be altered to any direction after uncovering the loss landscape of weight in private learning, as variants developed versions of SAM (Foret et al., 2020) are investigated in standard training (Zhuang et al., 2021; Kwon et al., 2021; Kim et al., 2023b; Andriushchenko & Flammarion, 2022), as discussed in Appendix E.4.

---

**Algorithm 1:** Weight Multiplicity

**Input:** Initial parameter $w_0$, multiplicity $K$, learning rate $\eta$, radius $\rho$, clipping threshold $C$,
   variance $\sigma^2$ from Proposition A.1, and small $\gamma > 0$ to prevent zero division.
**Output:** Final parameter $w_T$.
**for** $t = 1, 2, \ldots, T$ **do**
 Random sampling a private mini-batch $X_t^{pr}$ and a public mini-batch $X_t^{pub}$.
 **for** $k = 1, 2, \ldots, K$ **do**
  $v_t^k = \rho \frac{\nabla\ell(w_t; X_t^{pub_k})}{\|\nabla\ell(w_t; X_t^{pub_k}))\| + \gamma}$
 **end**
 **for** $i = 1, 2, \ldots, |X_t^{pr}|$ **do**
  **for** $k = 1, 2, \ldots, K$ **do**
   **if** *weight-augment multiplicity* **then**
    $g_t^{(i,k)} = \nabla\ell\left(w_t + v_t^k; \mathtt{aug}^k(x_i)\right)$
   **else if** *only augment multiplicity (De et al., 2022)* **then**
    $g_t^{(i,k)} = \nabla\ell\left(w_t; \mathtt{aug}^k(x_i)\right)$
  **end**
  $g_t^i = \frac{1}{K} \sum_{k=1}^K g_t^{(i,k)}$
 **end**
 $g_t^{pr} = \frac{1}{|X_t^{pr}|} \sum_{i=1}^{|X_t^{pr}|} \mathtt{clip}\left(g_t^i, C\right)$
 $\tilde{g}_t^{pr} = g_t^{pr} + \mathcal{N}(\mathbf{0}, C^2\sigma^2\mathbf{I})$
 $w_{t+1} = w_t - \eta\tilde{g}_t^{pr}$
**end**

---

## B PROOFS

### B.1 PROOF OF THEOREM 4.1.

Following (Karras et al., 2022), we address this theorem with a finite number of samples $S_{data} = \{\boldsymbol{x}_1, \cdots, \boldsymbol{x}_N\}$. As mentioned in Appendix B.3 of (Karras et al., 2022), $p_{data}$ and $p_{pub}$ can be represented by mixtures of Dirac delta distributions:

$$p_{data}(\boldsymbol{x}) = \frac{1}{N} \sum_{i=1}^{N} \delta(\boldsymbol{x} - \boldsymbol{x}_i),$$

and

$$p_{pub}(\boldsymbol{x}) = \frac{1}{n} \sum_{i=1}^{n} \delta(\boldsymbol{x} - \boldsymbol{x}_i),$$

where $\boldsymbol{x} \in \mathcal{X}$ which is discrete sample space, since any pixel of an image can be represented by int between 0 to 255. Then, by the definition of KL divergence with a discrete probability distribution,

$$
\begin{aligned}
D_{KL}(p_{data}\|p_\theta) &= \sum_{\boldsymbol{x}\in\mathcal{X}} \left( p_{data}(\boldsymbol{x}) \log \frac{p_{data}(\boldsymbol{x})}{p_\theta(\boldsymbol{x})} \right) \\
&= \sum_{\boldsymbol{x}\in S_{pub}} \left( p_{data}(\boldsymbol{x}) \left[ \log \frac{p_{pub}(\boldsymbol{x})}{p_\theta(\boldsymbol{x})} + \log \frac{p_{data}(\boldsymbol{x})}{p_{pub}(\boldsymbol{x})} \right] \right) + \sum_{\boldsymbol{x}\notin S_{pub}} \left( p_{data}(\boldsymbol{x}) \log \frac{p_{data}(\boldsymbol{x})}{p_\theta(\boldsymbol{x})} \right) \\
&= D_{KL}(p_{pub}\|p_\theta) + \frac{n}{N} \log \frac{n}{N} + \sum_{\boldsymbol{x}\notin S_{pub}} \left( p_{data}(\boldsymbol{x}) \log \frac{p_{data}(\boldsymbol{x})}{p_\theta(\boldsymbol{x})} \right) \\
&= D_{KL}(p_{pub}\|p_\theta) + \frac{n}{N} \log \frac{n}{N} + \frac{1}{N} \sum_{\boldsymbol{x}\in S_{pr}} \left( \log \frac{1}{N p_\theta(\boldsymbol{x})} \right) \\
&= D_{KL}(p_{pub}\|p_\theta) + \frac{n}{N} \log \frac{n}{N} + \frac{N-n}{N} \log \frac{1}{N} - \frac{N-n}{N} \log \left( \prod_{\boldsymbol{x}\in S_{pr}} p_\theta(\boldsymbol{x}) \right)^{\frac{1}{N-n}} \\
&\geq D_{KL}(p_{pub}\|p_\theta) + \frac{n}{N} \log \frac{n}{N} + \frac{N-n}{N} \log \frac{1}{N} - \frac{N-n}{N} \log \frac{1}{N-n} \\
&= D_{KL}(p_{pub}\|p_\theta) + \frac{n}{N} \log \frac{n}{N} + \frac{N-n}{N} \log \frac{N-n}{N}.
\end{aligned}
$$

The inequality is followed by AM-GM inequality and the fact that the sum of probabilities is less than or equal to 1. Moreover, to make KL divergence well-defined in the finite setting, $p_\theta(\boldsymbol{x}) \neq 0$ where $p_{data}(\boldsymbol{x}) \neq 0$. Then, by the definition of $\tau_{pr} = \{(N-n) \min_{\boldsymbol{x}\in S_{pr}} p_\theta(\boldsymbol{x})\}^{-1}$,

$$
\begin{aligned}
D_{KL}(p_{data}\|p_\theta) &= D_{KL}(p_{pub}\|p_\theta) + \frac{n}{N} \log \frac{n}{N} + \frac{1}{N} \sum_{\boldsymbol{x}\in S_{pr}} \left( \log \frac{1}{N p_\theta(\boldsymbol{x})} \right) \\
&\leq D_{KL}(p_{pub}\|p_\theta) + \frac{n}{N} \log \frac{n}{N} + \frac{N-n}{N} \left( \log \frac{(N-n)\tau_{pr}}{N} \right) \\
&= D_{KL}(p_{pub}\|p_\theta) + \frac{n}{N} \log \frac{n}{N} + \frac{N-n}{N} \log \frac{N-n}{N} + \frac{N-n}{N} \log \tau_{pr}.
\end{aligned}
$$

Note that as $n \to N$, $p_{pub} \to p_{data}$, and above two inequality achieves equality. Moreover, as $|S_{pr}| = N - n$, we have $\min p_\theta(\boldsymbol{x}) \leq \frac{1}{N-n}$, ensuring that $\log \tau_{pr} \geq 0$.

This theorem tells that we can approximate $p_{data}$ with $p_{pub}$ with in-distribution $S_{pub}$. Furthermore, when $n$ small, we can approximate $p_{pr}$ with $p_{pub}$ since $S_{data}$ is similar to $S_{pr}$.

## B.2 PROOF OF THEOREM 5.1.

Motivated by the Theorem 1 of (Gong et al., 2021),

$$\frac{1}{K}\sum_{k=1}^{K}\ell(\boldsymbol{w}+\boldsymbol{v}^k;\operatorname{aug}^k(\boldsymbol{x}_i)) = \frac{1}{K}\sum_{k=1}^{K}\{\ell(\boldsymbol{w};\operatorname{aug}^k(\boldsymbol{x}_i)) + \nabla_{\boldsymbol{w}}\ell(\boldsymbol{w};\operatorname{aug}^k(\boldsymbol{x}_i))^T\boldsymbol{v}^k + O(\rho^2)\} \quad (8)$$

$$= \frac{1}{K}\sum_{k=1}^{K}\{\ell(\boldsymbol{w};\operatorname{aug}^k(\boldsymbol{x}_i)) + \rho\|\nabla_{\boldsymbol{w}}\ell(\boldsymbol{w};\operatorname{aug}^k(\boldsymbol{x}_i))\|\cos\theta^k\} + O(\rho^2) \quad (9)$$

$$= \ell(\boldsymbol{w};\boldsymbol{x}_i) + \frac{1}{K}\sum_{k=1}^{K}\{\ell(\boldsymbol{w};\operatorname{aug}^k(\boldsymbol{x}_i)) - \ell(\boldsymbol{w};\boldsymbol{x}_i)\} \quad (10)$$

$$+ \Phi(\boldsymbol{w},\boldsymbol{v}^1:\boldsymbol{v}^K;\boldsymbol{x}_i) + O(\rho^2) \quad (11)$$

$$= \ell(\boldsymbol{w};\boldsymbol{x}_i) + \frac{1}{K}\nabla_{\boldsymbol{x}}\ell(\boldsymbol{w};\boldsymbol{x}_i)^T\sum_{k=1}^{K}(\operatorname{aug}^k(\boldsymbol{x}_i) - \boldsymbol{x}_i) + O(h^2) \quad (12)$$

$$+ \Phi(\boldsymbol{w},\boldsymbol{v}^1:\boldsymbol{v}^K;\boldsymbol{x}_i) + O(\rho^2) \quad (13)$$

$$= \ell(\boldsymbol{w};\boldsymbol{x}_i) + \Phi(\boldsymbol{w},\boldsymbol{v}^1:\boldsymbol{v}^K;\boldsymbol{x}_i) + \Psi(\boldsymbol{w};\boldsymbol{x}_i) + O(\rho^2) + O(h^2) \quad (14)$$

We now explain the techniques in the proof.

Equation (8) and Equation (12) are followed by Taylor expansion. $\theta^k$, defined in Equation (9), is the angle between the gradient $\nabla_{\boldsymbol{x}}\ell(\boldsymbol{w};\operatorname{aug}(\boldsymbol{x}_i))$ and $\boldsymbol{v}^k$, which is always positive since $\boldsymbol{v}^k$ is loss-ascending direction. Therefore, by defining $\Phi(\boldsymbol{w},\boldsymbol{v}^1:\boldsymbol{v}^K;\boldsymbol{x}_i)$ as Equation (11), we can get

$$\frac{\rho}{K}\min_k\cos\theta^k\sum_{k=1}^{K}\|\nabla_{\boldsymbol{w}}\ell(\boldsymbol{w};\operatorname{aug}^k(\boldsymbol{x}_i))\| \leq \Phi(\boldsymbol{w},\boldsymbol{v}^1:\boldsymbol{v}^K;\boldsymbol{x}_i) \leq \frac{\rho}{K}\sum_{k=1}^{K}\|\nabla_{\boldsymbol{w}}\ell(\boldsymbol{w};\operatorname{aug}^k(\boldsymbol{x}_i))\|,$$

so that $c_1 = \frac{\rho}{K}\min_k\cos\theta^k$ and $c_2 = \frac{\rho}{K}$. Note that the higher cosine similarity between $\boldsymbol{v}_k$ and gradient ensures the larger $c_1$, providing the higher weight to the weight regularization term. Moreover,

$$\Psi(\boldsymbol{w};\boldsymbol{x}_i) = \frac{1}{K}\nabla_{\boldsymbol{x}}\ell(\boldsymbol{w};\boldsymbol{x}_i)^T\sum_{k=1}^{K}(\operatorname{aug}^k(\boldsymbol{x}_i) - \boldsymbol{x}_i) = \|\nabla_{\boldsymbol{x}}\ell(\boldsymbol{w};\boldsymbol{x}_i)\|\frac{1}{K}\sum_{k=1}^{K}\alpha^T(\operatorname{aug}^k(\boldsymbol{x}_i) - \boldsymbol{x}_i),$$

where $\alpha = \nabla_{\boldsymbol{x}}\ell(\boldsymbol{w};\boldsymbol{x}_i)/\|\nabla_{\boldsymbol{x}}\ell(\boldsymbol{w};\boldsymbol{x}_i)\|$. By assumption, $\ell(\boldsymbol{w};\operatorname{aug}^k(\boldsymbol{x}_i)) \geq \ell(\boldsymbol{w};\boldsymbol{x}_i)$, then

$$c = \frac{1}{K}\sum_{k=1}^{K}\alpha^T(\operatorname{aug}^k(\boldsymbol{x}_i) - \boldsymbol{x}_i) \geq \min_k(\alpha^T(\operatorname{aug}^k(\boldsymbol{x}_i) - \boldsymbol{x}_i)) = c_3 \geq 0.$$

Therefore, we got $c \leq \frac{1}{K}\sum_{k=1}^{K}\|\operatorname{aug}^k(\boldsymbol{x}_i) - \boldsymbol{x}_i\| = c_4$.

## C  EXPERIMENTAL SETTINGS

### C.1  CLASSIFICATION

**Private training**  For private learning, we adopt all the techniques of (De et al., 2022) with WRN-16-4. We employ the techniques such as *augmentation multiplicity* to minimize the averaged loss of various augmentations $\mathcal{L}_i(\boldsymbol{w}) = \mathbb{E}_k[\ell(\boldsymbol{w}, \text{aug}^k(\boldsymbol{x}_i))]$, weight standardization (Qiao et al., 2019), and Exponential Moving Average (EMA). We re-implement the JAX official code of (De et al., 2022) in https://github.com/google-deepmind/jax_privacy and extended methods using Pytorch Opacus (Yousefpour et al., 2021) libraries.

We present the experimental details for CIFAR-10 in Table 10 and CIFAR-100 in Table 11 with their search spaces and best hyperparameter values. All experiments are conducted with DP-SGD with momentum 0 unless otherwise specified. As private learning is hugely affected by the hyperparameter settings, we use a different search space for cold and warm settings. We calculated the noise level $\sigma$ for training with the hyperparameters in Tables 10 and 11 using Opacus libraries.

Table 10: Hyperparameters for CIFAR-10.

| Setup | Hyper-parameter | Search space | Best values | | | | |
|---|---|---|---|---|---|---|---|
| | $\varepsilon$ | $\{1, 2, 3, 4, 6\}$ | 1 | 2 | 3 | 4 | 6 |
| | $\delta$ | $\{10^{-5}\}$ | $10^{-5}$ | $10^{-5}$ | $10^{-5}$ | $10^{-5}$ | $10^{-5}$ |
| | Multiplicity $K$ | $\{16\}$ | 16 | 16 | 16 | 16 | 16 |
| Warm | Batch size | $\{4096\}$ | 4096 | 4096 | 4096 | 4096 | 4096 |
| | Clipping norm $C$ | $\{1\}$ | 1 | 1 | 1 | 1 | 1 |
| | Epochs | $\{15, 20, 30, 40\}$ | 15 | 30 | 30 | 30 | 20 |
| | Learning rate $\eta$ | $\{0.1, 0.5, 1, 2, 4\}$ | 0.5 | 0.5 | 1 | 1 | 2 |
| + Extended | Public batch size | $\{32, 64, 128\}$ | 64 | 64 | 64 | 64 | 64 |
| | Weight multiplicity $\rho$ | $\{0.05, 0.1, 0.2\}$ | 0.05 | 0.1 | 0.1 | 0.1 | 0.1 |

Table 11: Hyperparameters for CIFAR-100.

| Setup | Hyper-parameter | Search space | Best values | | | |
|---|---|---|---|---|---|---|
| | $\varepsilon$ | $\{1, 2, 6, 10\}$ | 1 | 2 | 6 | 10 |
| | $\delta$ | $\{10^{-5}\}$ | $10^{-5}$ | $10^{-5}$ | $10^{-5}$ | $10^{-5}$ |
| | Multiplicity $K$ | $\{16\}$ | 16 | 16 | 16 | 16 |
| Warm | Batch size | $\{4096\}$ | 4096 | 4096 | 4096 | 4096 |
| | Clipping norm $C$ | $\{1\}$ | 1 | 1 | 1 | 1 |
| | Epochs | $\{25, 50, 75\}$ | 25 | 50 | 75 | 75 |
| | Learning rate $\eta$ | $\{0.1, 0.5, 1, 2, 4\}$ | 0.5 | 0.5 | 1 | 1 |
| + Extended | Public batch size | $\{32, 64, 128\}$ | 64 | 64 | 64 | 64 |
| | Weight multiplicity $\rho$ | $\{0.05, 0.1, 0.2\}$ | 0.1 | 0.1 | 0.05 | 0.05 |

**Warm-up training**  We present the experimental details of the warm-up phase with SGD (not DP-SGD) in Table 12. We use SGD with momentum 0.9 as a default setting for warm-up training. For the warm setting, we train epochs until convergence since the number of training data of 2,000 is less than the warmSyn of 50,000 samples.

**Pre-trained model**  We use pre-trained Vision Transformers such as DeiT (Touvron et al., 2021), and CrossViT (Chen et al., 2021). We used the ghost clipping methods proposed in (Bu et al., 2022) and their GitHub code from https://github.com/woodyx218/private_vision. For the cold setting, we trained the models with 5 epochs with Adam optimizer with a learning rate of 0.002. We used a batch size of 1,000 following the default settings in GitHub. We tested various ranges of model sizes. For the warm settings, we took a grid search on the learning rate of $\{0.0005, 0.001, 0.002\}$ on both the warm-up phase and the private training phase.

Table 12: Hyperparameters for warm-up phase on CIFAR-10 and CIFAR-100.

| Hyper-parameter | Search space | CIFAR-10 | CIFAR-100 |
|---|---|---|---|
| Batch size | {64, 100, 128, 256} | 100 | 64 |
| Epochs | {50, 100, 200} | 100 | 200 |
| Learning rate $\eta$ | {0.05, 0.1, 0.2, 0.3} | 0.05 | 0.1 |
| Momentum | {0.9} | 0.9 | 0.9 |
| Learning rate decay | {Cosine} | Cosine | Cosine |
| Weight decay | {$5 \times 10^{-4}$} | $5 \times 10^{-4}$ | $5 \times 10^{-4}$ |
| Radius $\rho$ for SAM | {0.05, 0.1, 0.2} | 0.1 | 0.1 |

### C.2 DIFFUSION SYNTHESIS

**EDM settings** We implemented EDM (Karras et al., 2022) from their official GitHub code from `https://github.com/NVlabs/edm` and DG (Kim et al., 2023a) from their official GitHub code from `https://github.com/alsdudrla10/DG`. We trained the EDM model using the base settings as reported on the official GitHub repository. For the CIFAR-10 dataset, we utilized a batch size of 512 images, distributed among four NVIDIA GeForce RTX 3090 GPUs, while maintaining the other setting as the default setting. Specifically, we use a learning rate of $10^{-3}$, an EMA coefficient of 0.5, duration of 200. The detailed settings are reported in Table 7 of (Karras et al., 2022). We sampled images with $\sigma_{min} = 0.002, \sigma_{max} = 80, \rho = 7, S_{churn} = 0, S_{min} = 0, S_{max} = \infty, S_{noise} = 1$, and a step size of 18, as the default setting. For the CIFAR-100 dataset, we employed a batch size of 1024 images, distributed among four NVIDIA A100 GPUs, while maintaining the same settings as those used for the CIFAR-10 dataset. The training took less than 3 days with four NVIDIA GeForce RTX 3090 GPUs. Image sampling with a step size of 18 and a batch size of 500, took less than 30 seconds per batch when using a single NVIDIA GeForce RTX 3090 GPU. Therefore, it took about one hour to sampling 50K images.

**DG settings** We trained both the classifier and discriminator for DG from scratch with synthetic data from the 2,000 public data. We avoided using pre-trained models from the CIFAR or Imagenet datasets to solely investigate the effects by using in-sample data.

## D ADDITIONAL NOTES

### D.1 DEEPER ANALYSIS FOR TOY EXAMPLE IN SECTION 4.1

For the spiral dataset, the radius increases proportionally with the angle and at each location, the points have a probability distribution that decreases proportionally with the cumulative sum. In other words, points closer to the origin have higher probabilities. With this dataset, we construct a simple diffusion model with time step 20 and two diffusion blocks, containing a linear layer with unit 64. We train the model 10,000 epochs with a learning rate of 0.001 using Adam optimizer. We train SVM classifiers with an RBF kernel, setting the hyperparameter C to 1,000 to enforce a hard margin. All other settings remained at their default values, following the conventions of the sklearn library. The diffusion models closely approximate the true data distribution, as shown in Figure 1b. The diffusion model effectively approximated the distribution, regardless of the number of training data in region A. However, in region C, where the true density is high, the generated distribution shows a higher probability when the number of training data is limited. This indicates that when the number of training data is small, the model tends to memorize the training dataset. Conversely, in regions B and D, where the true density is low, the probability distribution of generated samples is lower than the true distribution, even the true density is low. This indicates that when the number of samples is small, the model tends to ignore the tail distribution. As the number of in-distribution data is small, we should be aware of using synthetic data in terms of data memorization and ignore the tail part.

**GAN mode collapse** The diversity problem is more significant in other generative models, such as generative adversarial networks (GAN) (Goodfellow et al., 2020). As GAN can generative images with high fidelity, however, GAN suffers from training instability to control both generator and

discriminatory, sometimes generating only a small portion of data repeatedly called mode collapse. The mode collapse happens when the convergence speed of the discrimination is faster than that of the generator, which induces the generator to generate the same images which can confuse the discriminator without generating similar images to real data. Figure 5 illustrates that the mode collapse is easy to happens with a smaller number of samples on Ring data. Thus, we mainly focus on diffusion models in this paper.

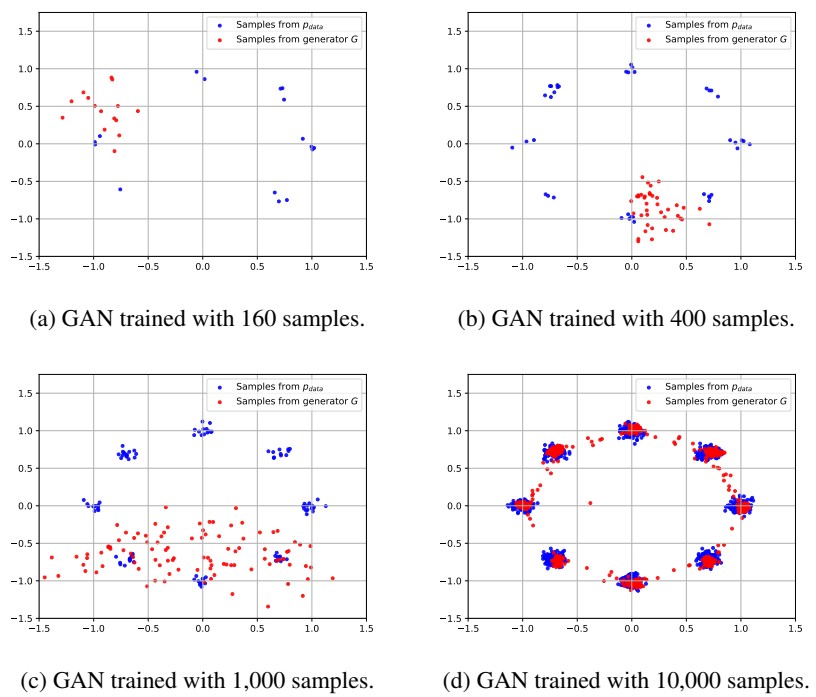

(a) GAN trained with 160 samples.

(b) GAN trained with 400 samples.

(c) GAN trained with 1,000 samples.

(d) GAN trained with 10,000 samples.

Figure 5: Mode collapse of GAN with a different number of samples on Ring data.

## D.2 DETAILS OF DISCRIMINATOR GUIDANCE

EDM (Karras et al., 2022) can achieve better generation than DDPM (Ho et al., 2020) by a higher-order Runge-Kutta method for sampling process and a new stochastic sampler based on $\sigma$. Thus, DG (Kim et al., 2023a) used the EDM as a base model for generating CIFAR datasets. To control the trade-off between fidelity and diversity in diffusion models, Dhariwal & Nichol (2021) suggested using the classifier to diffusion networks. The classifier, which is trained on noisy images during diffusion steps and their labels, can force the model to generate certain types of images based on their labels. To push further, DG (Kim et al., 2023a) adopted another network called discriminator. The discriminator is trained to decide whether the images during the diffusion process are generated from real data or not. Thus, similar to the discriminator in GAN, the model can force the diffusion model to generate more similar images to the real datasets. Both studies enable the users to control the level of fidelity and diversity in diffusion sampling, where the optimal FID is obtained with a moderate level of fidelity and diversity. Instead, we focus on a high level of weight in the discriminator to generate images with high diversity, rather than repeating typical images with high fidelity.

## D.3 TRAINING DYNAMICS

For detailed analysis in Figure 3, we show additional results representing the learning dynamics of the private training phase. We measure the gradient norm and loss of private and synthetic data, as shown in Figure 6. As DP-SGD focuses on private data, the loss of private data decreases constantly, while the loss of public data is increasing. On the other hand, DOPE-SGD controls the loss of both synthetic and private data. However, in a certain range of training, DOPE-SGD occasionally fails to converge and diverge, where their gradient norms explode and the accuracy plummets to 0. As

private training with DP-SGD usually uses a larger learning rate than of standard training, explicitly minimizing the public gradient might be dangerous as the model can be overfitted to training data and stuck into sharp minima during training. However, the proposed weight multiplicity shows the intermediate behavior of DP-SGD and DOPE-SGD.

Interestingly, Figure 6d illustrates the gradient norm, the same as Figure 3b but without clipping. Both DOPE-SGD and DP-SGD show diverging gradient norms during training, where weight multiply can decrease the norm even without clipping. This might be a main reason to explain the usefulness of weight multiplicity.

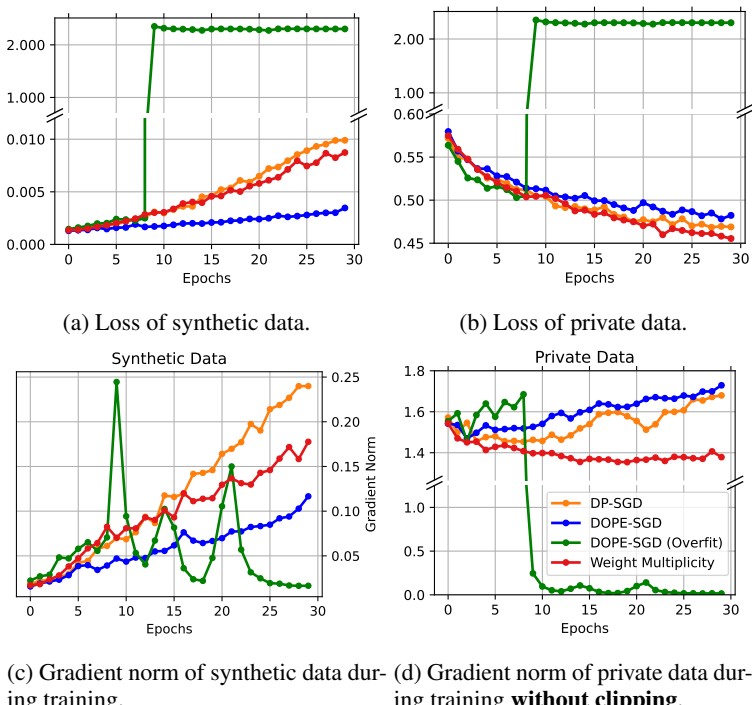

(a) Loss of synthetic data.

(b) Loss of private data.

(c) Gradient norm of synthetic data during training.

(d) Gradient norm of private data during training **without clipping**.

Figure 6: Learning dynamics during private training of various optimization methods.

## D.4 CLUSTER OF GRADIENTS IN THE WEIGHT SUB-SPACE

It is known that SGD happens in tiny subspaces among large weight dimension, which usually has the same dimension as the class number in classification tasks (Papyan, 2019). We can find a meaningful manifold of the weight space by decomposing the weight matrix and projecting to the space spanned by the top eigenvectors of the loss-weight function (Sagun et al., 2018; Lee et al., 2023). Various researchers have already used the decomposition and low-rank approximation to mitigate the effect of clipping and noise addition in private learning (Yu et al., 2021b; Ye & Shokri, 2022).

Similar to Figure 4, we provide the clusters of individual gradients projected onto the 2D plane spanned by the two mean gradients with normalization in Figure 7. Figures 7a and 7b show similar results of Figure 4 with different classes, where individual gradient directions of both datasets are similar among intra-class samples but differ between classes. Figures 7d 7e, and 7f illustrate the difference between private and synthetic data when projected with their mean vectors. The difference in gradients between private and synthetic data is small. However, their gradient norm is significantly different as shown in Figure 7c, where the norm of synthetic data is near the origin but the norm of a private vector is relatively large along their direction.

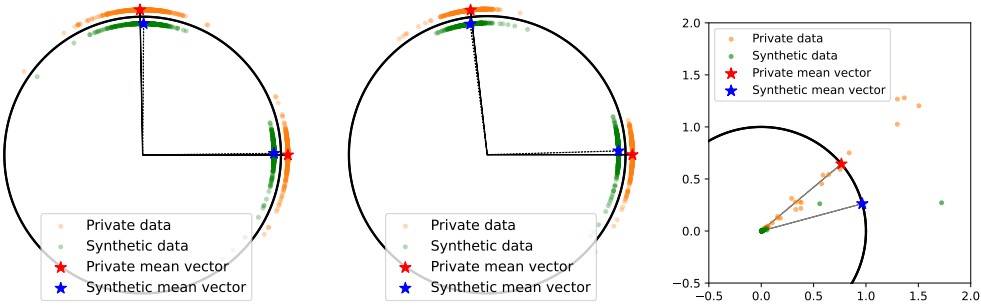

(a) Within/between-class similarity of Dog and Truck. (b) Within/between-class similarity of Truck and Bird. (c) Similarity without normalization of Automobile class.

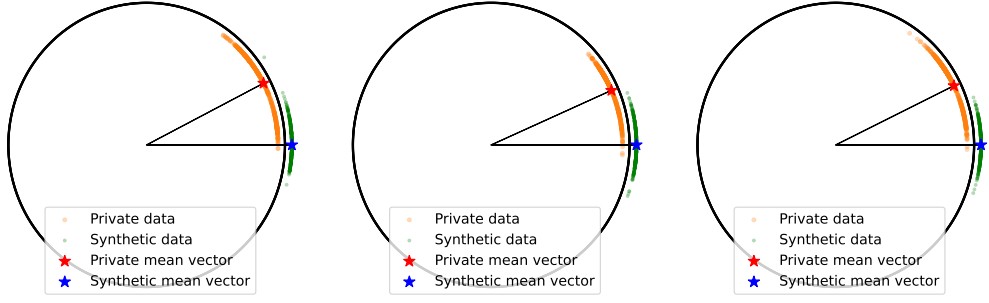

(d) Similarity of private and synthetic data of Dog class. (e) Similarity of private and synthetic data of Truck class. (f) Similarity of private and synthetic data of Bird class.

Figure 7: Clusters of individual gradients projected onto the 2D plane spanned by the two mean gradients.

# E    ABLATION STUDY

## E.1    DG AFTER AUGMENTATION

The CAS for applying common + cutout augmentation to the synthetic images generated by EDM + DG are presented in Table 13. Interestingly, the combination of DG and augmentation does not show significant performance improvement than images with $w_d = 0$. The performance becomes worse in larger $w_d$, as the image quality is decreased. We believe that the deficient diversity in sparse in-sample data is covered with augmentation techniques and leave their combination for the future work. The lower accuracy of both the classifier and discriminator of DG trained with 2,000 samples and limited synthetic images might be a reason. Note that Wang et al. (2023) chose the synthetic images with FID for improving performance in adversarial training.

Table 13: CAS comparison of synthetic data generated from EDM and DG ($w_d = 3, 10, 20, 30$) with common + cutout augmentation on CIFAR-10.

| Sampling | EDM | + DG ($w_d$=3) | + DG ($w_d$=10) | + DG ($w_d$=20) | + DG ($w_d$=30) |
|---|---|---|---|---|---|
| CAS (%) | 80.40 | 80.44 | 80.62 | 79.69 | 78.89 |

## E.2    DIFFERENT PRIVACY BUDGET $\varepsilon$ ON PRE-TRAINED CIFAR-100

We additionally append the experimental results, using the same hyperparameter setup of Table 7, in Table 14 with privacy budget of $\varepsilon \in \{1, 4\}$.

Table 14: Test accuracy (%) of private classification using pre-trained models.

| Datasets | Privacy budget | | $\varepsilon = 1$ | | | $\varepsilon = 4$ | | |
|---|---|---|---|---|---|---|---|---|
| | Architecture | Pre-trained | Cold | Warm | Ours | Cold | Warm | Ours |
| CIFAR -100 | CrossViT small 240 (26.3M) | ✓ | 66.88 | 74.70 | **78.91** | 72.91 | 78.80 | **81.75** |
| | CrossViT 18 240 (42.6M) | ✓ | 71.27 | 78.60 | **81.39** | 76.22 | 81.73 | **83.75** |
| | DeiT base patch16 224 (85.8M) | ✓ | 63.84 | 81.55 | **81.62** | 72.84 | **84.60** | 84.43 |
| | CrossViT base 240 (103.9M) | ✓ | 71.08 | 77.30 | **79.65** | 76.63 | 80.43 | **82.63** |

### E.3 EFFECT OF RADIUS $\rho$ ON WEIGHT MULTIPLICITY

We take an ablation study on the effect of radius $\rho$ in Table 15. We test the effects on CIFAR-10 datasets while maintaining all the other settings the same.

Table 15: Performance of the weight multiplicity on different radius $\rho$ on CIFAR-10.

| Radius $\rho$ | $\varepsilon = 2$ | $\varepsilon = 4$ | $\varepsilon = 6$ |
|---|---|---|---|
| 0.05 | 85.21 | 86.53 | 87.01 |
| 0.1 | 85.78 | 86.59 | 87.23 |
| 0.2 | 85.17 | 86.51 | 85.38 |

### E.4 ADDITIONAL OPTIMIZATION METHODS

We try to use various techniques for private learning to improve classification performance. The classification results with $(2, 10^{-5})$-DP are presented in Table 16.

Table 16: Test accuracy of adversarial training and adaptive weight multiplicity on CIFAR-10.

| Datasets | Architecture | Public | Synthesis | Methods | $\varepsilon = 2$ |
|---|---|---|---|---|---|
| CIFAR -10 | WRN-16-4 (2.74M) | ✓ | EDM | Adversarial training $(8/255)$ in warm-up | 77.25 |
| | | ✓ | EDM | Adversarial training $(2/255)$ in warm-up | 81.55 |
| | | ✓ | EDM | Adaptive weight multiplicity | 85.16 |

**Adversarial Training** Adversarial training (Madry et al., 2018; Zhang et al., 2019) aims to make decision boundaries smooth in terms of input space. Furthermore, the importance of generative models is well under-studied in adversarial training (Gowal et al., 2021; Rebuffi et al., 2021; Wang et al., 2023). Similar to DP training, the task of adversarial training is harder than standard training, Wang et al. (2023) observed that utilizing the synthetic data with EDM can improve the generalization performance and prevent robust overfitting without extra data samples. Therefore, we adopt to use of adversarial training in the warm-up phase. However, adversarial training methods are designed to reduce the accuracy of PGD-10 (Madry et al., 2018), and the standard accuracy is significantly dropped. Therefore, the private classification results are also decreased.

**Different direction of perturbation in weight multiplicity** As mentioned in Section 5.2, the selection of $v$ in the parameter space is controversial even in standard training (Andriushchenko & Flammarion, 2022; Kim et al., 2023b; Kwon et al., 2021; Zhuang et al., 2021). Therefore, we adopted some techniques into Equation (6) to check the methods are still working on the private training. We mainly adopt adaptive SAM (ASAM) (Kwon et al., 2021) to better select the perturbation $v$. ASAM poses more weight to the layers that have a larger weight norm, in contrast to SAM which has the same importance along all the layers. However, it requires one additional controllable hyper-parameter than SAM, thus hard to tune in our experimental settings.

## F GENERATED IMAGES

We illustrate the samples of generated images using in-sample public data. Figure 8a demonstrates the memorization within CIFAR-10, by random sampling synthetic images in the first row and choose the nearest samples in the synthetic dataset. Figure 8b and Figure 8c are obtained from generated images of CIFAR-10, and Figure 8d is obtained from generated images of CIFAR-100.

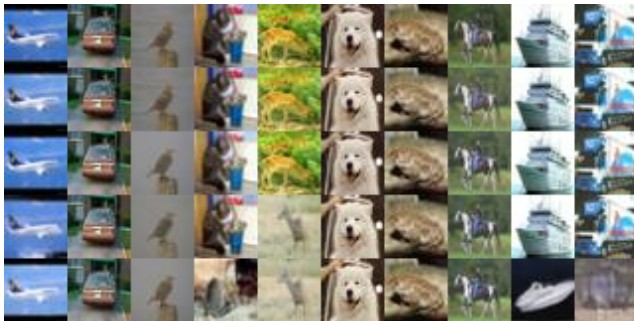

(a) Illustration of memorization within generated CIFAR-10 images.

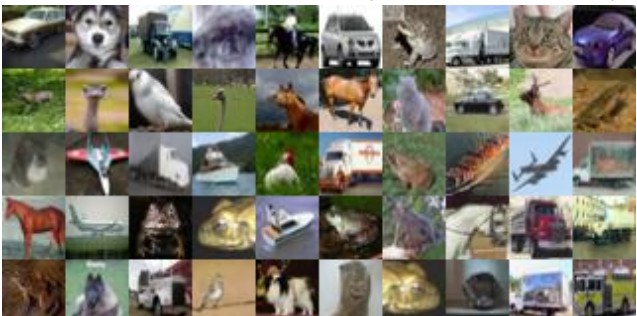

(b) Samples of generated CIFAR-10 images with EDM using 2,000 samples.

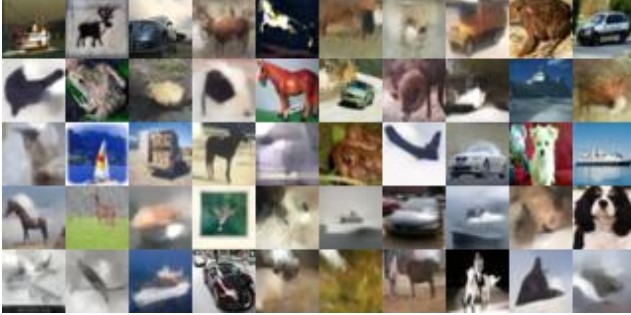

(c) Samples of generated CIFAR-10 images with EDM + DG ($w_d$=30) using 2,000 samples.

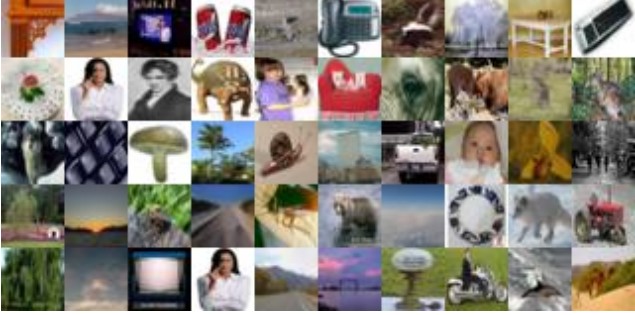

(d) Samples of generated CIFAR-100 images with EDM using 2,000 samples.

Figure 8: Samples of generated images using in-sample public data.

