# OpenReview forum: "Improving Private Training via In-distribution Public Data Synthesis and Generalization"
_ICLR.cc/2024/Conference — ICLR 2024 Conference Withdrawn Submission_

### Official Review · Reviewer_jmD4 · 2023-10-24

**Soundness:** 3 good
**Presentation:** 4 excellent
**Contribution:** 3 good
**Rating:** 6
**Confidence:** 3

**Summary:**

The authors have conducted a thorough empirical study on how to improve the utility of DP image classifiers with a limited amount of public data. Importantly, the public data share the same distribution as the private data. The findings are that (1) using diffusion models and many augmentation techniques to augment the public information, (2) and making the optimization smoother can significantly improve the utility of the DP image classifier.

**Strengths:**

1. The paper is clearly written, with all informative figures and tables. The method is overall well-motivated.
2. The authors have done a nice ablation study on each component of the method.
3. The experiment results show the effectiveness of their method compared to baselines. Improving the privacy-utility tradeoff is always of great interest to both researchers and engineers in the DP community.

Self-claim: I am not an expert in this particular direction, so I am not too certain how significant these empirical findings are compared to prior works.

**Weaknesses:**

1. This work to me seems to be an empirical study (with great completeness and thoroughness though). The technical novelty is limited. There are a few key findings/tricks to improve the utility, namely, (1) using a limited amount of in-distribution public data; (2) using diffusion models to synthesize more public images; (3) using a few augmentation techniques to further enhance the diversity; (4) taking a smoother step in DP-SGD. However,
    + Trick (1)(2)(3) are all known techniques. They are not just used in the non-private training but also in the private training. I do not find too much contribution towards these training tricks.
    + Trick (4) is more interesting and contains two parts: SAM (to mitigate overfitting), and weight multiplicity (to smoothen the optimization). However, SAM is entirely borrowed from prior work, and weight multiplicity did an incremental revision to another existing technique called augmentation multiplicity. I wish the authors could clarify if there are more novelties, and I also suggest the authors write an explicit paragraph summarizing the contributions of this work.

2. The motivation of adding a perturbation vector $v$ in weight multiplicity is unclear.  What is the intuitive meaning of this vector and why does it contribute to smoother loss landscape? I also expect to see the comparison of weight and augmentation multiplicity.

3. Results in table 5 look very nice, but I am confused why the authors use EDM for their method but using DDPM for other baselines. If EDM is better, it should be used for all baselines to indicate the effectiveness of optimization tricks.

**Questions:**

See the weakness above.

---

### Official Review · Reviewer_9BFG · 2023-11-01

**Soundness:** 2 fair
**Presentation:** 2 fair
**Contribution:** 2 fair
**Rating:** 3
**Confidence:** 3

**Summary:**

The paper studies privacy-preserving deep model learning with a small amount of public data. It follows a previous framework: learning to generate synthetic data from public data, pre-training the model from synthetic data + public data, and running a variant of DP-SGD to maximize the usage of public data. In detail, it proposes to use EDM instead of DDPM in the baseline to generate synthetic data and leverage the SAM and weight multiplicity in the optimization. Then it evaluate the proposed methods and baselines on both CIFAR10 and CIFAR100 and demonstrate the strengths of the proposed algorithm.

**Strengths:**

1. The studied problem is important: how to improve the trade-off between privacy and accuracy when training a deep learning model.
2. The results are promising. For example, the proposed method can achieve 85.78% when $\varepsilon=2.0$ on the CIFAR-10 dataset while the best baseline can only achieve 75.1%.

**Weaknesses:**

It is not well explained when introducing each module in the method:
- It is not clear what is the conclusion from Theorem 4.1. What does "when $r_1$ is small, ..., can approximate $p_{data}$ and $p_{pr}$" mean? It is also not clear the EDM is adopted from Theorem 4.1.
- To mitigating overfitting to public data, why not directly apply SAM in the warm-up phase (pretraining optimization)?
- If multiplicity is an approximation of gradient norm regularizer, why not directly apply gradient norm regularizer? What's the additional benefit beyond just the regularizer?

**Questions:**

In the proof of Theorem 4.1: why is the third equation (i.e. $\sum_{x\in S_{sub}}p_{data}(x)\log\frac{p_{pub}(x)}{p_{\theta}(x)}=D_{KL}(p_{pub}||p_{\theta})$ ) true?

Please also check the "Weaknesses".

---

### Official Review · Reviewer_8NJG · 2023-11-02

**Soundness:** 2 fair
**Presentation:** 2 fair
**Contribution:** 2 fair
**Rating:** 5
**Confidence:** 4

**Summary:**

The core problem addressed in this paper is differentially private image classification with access to in-distribution public data. The paper proposes a range of modifications to generate useful synthetic data which is then used as the pre-training data. In addition the paper proposes using sharpness aware minimisation style optimisation for both public and private data. As a result, the paper shows gains in performance on CIFAR-10 and CIFAR-100.

**Strengths:**

* There are a range of ideas proposed in this paper for improving diffusion model based image generation as well as private optimisation.
* The biggest strength of the paper is what appears to be good test accuracy on CIFAR-10 and CIFAR-100.
* Exploring the extent to which improving image generation can help in DP image classification is an interesting work as the amount of available public data in sensitive domains is expected to be less.

**Weaknesses:**

* __W1__ The clarity of the paper is severely lacking at the current stage. Some major ones are as follows:
    * In Theorem 4.1, it is not clear from the theorem what the distributions $p_{\text{data}},p_{\text{pub}},p_{\text{pr}}$ are and how they are defined whereas they are the main objects whose properties are being bounded. In the same theorem, there is a mention of trained model, that generates a distribution $p_{\theta}$. But the model is not defined and it is not clear what kind of distribution this is. More importantly, it is not clear what the significance of the theorem is and how it improves over Nasr et. al. 2023 which appears to be the main motivation for writing this. Therefore I did not understand what this section is contribution and I think it needs to be rewritten with more clarity.
    * Similarly, in Theorem 5.1, the same problems hold. There are a lot of notations that are not defined at all and it is not clear what this is contributing. I would recommend distilling this into a self-contained form with less notation and if necessary a more informal version for the main paper.
    * The introduction is also not very communicative of the problem the paper is trying to solve and motivate the approach takes. For example, 1) why is public data going to be used multiple times ? 2) what is geometric-based optimisation and why should it help overfitting ? 3) why is overfitting with respect to the public data bad if we only care about generalisation of the private train error to the test set (as that is what we are doing in DP ERM)
     * Eq 2 is also not correct. Its not possible to add a vector (gradient) to a distribution ( $\mathcal{N}(\ldots)$)

* __W2__ Regarding the soundness of the paper, I think various points either need to be clarified or corrected. Here are some examples I found.
     * Section 2.1 - "As Private training often exhibits training......uncovering smooth or flat minima is effective for generalization" - It is well known that DP implies generalization. The problem in DP training is not overfitting/lack of memorization but rather optimization.  And this is a problem here as this seems to motivate the whole contribution of `weight multiplicity' in this paper. I am confused regarding why this method is being introduced.
    * In Section 4.1, what SVM is being trained here ? The decision boundaries are highly non-linear for these SVM, so it cannot be linear. How is the non-linearity parameter chosen for the different figures in Figure 1 ?
    * Section 4.3 - if I understand correctly, using DG requires choosing a trade-off between fidelity and quality. But measuring these quantities need access to the private data ? Doesn't this result in a leakage of privacy ? I would be grateful if the authors would correct me here if I am wrong.
    * More generally, the paper proposes  a lot of different modifications that act on top of each other and the only component that comes with privacy guarantee is the vanilla DP-SGD. I think the paper needs a thorough discussion of the privacy guarantee of the whole method and possibly leakages. With so many moving parts, various things can leak privacy, see [1] for an exemplary case.

* __W3__ Significance: The experimental results on CIFAR 10/100 appear to be good.
    * However, given the empirical focus of the paper I think it needs  a more thorough experimentation to actually show its benefits more rigorously. Im particular, the paper should focus on datasets that are 1) used in De et. al. apart from CIFAR e.g. ImageNet 2) datasets far away from CIFAR e.g. high dimensional medical vision datasets like Patch-Camelyon (not necessarily this, but the authors can look at recent works to see what medical datasets are used in DP works e.g. see [2,3] )
    * Two relevant works that also use public indistribution data are GEP and PILLAR [2] but they don't use their gradients similar to Li et. al. or Asi et. al. Discuss GEP and PILLAR and compare with them. It may also be fair to use ImageNet Pretrained models from a privacy perspective as out-of-distribution public data.
    * Another important ablation to consider is distribution shift between public and private data. Can the authors show how the performance changes. In addition, I think the paper should also conduct an ablation with respect to the amount of available public data ?

[1] Mironov, Ilya. "On significance of the least significant bits for differential privacy." Proceedings of the 2012 ACM conference on Computer and communications security. 2012.
[2] Pinto, Francesco, et al. "PILLAR: How to make semi-private learning more effective." TPDP (2023).
[3] Berrada, Leonard, et al. "Unlocking Accuracy and Fairness in Differentially Private Image Classification." arXiv preprint (2023).

**Questions:**

I have listed the weaknesses in individual bullet points and each bullet point poses a question. I would be grateful for answer to those questions.

Of course I do not expect the authors to conduct the additional experiments in this rebuttal period but I believe they are needed for the paper to be published (unless I am convinced otherwise)

---

### Official Review · Reviewer_nRz4 · 2023-11-07

**Soundness:** 3 good
**Presentation:** 2 fair
**Contribution:** 3 good
**Rating:** 5
**Confidence:** 4

**Summary:**

The authors proposed a recipe of private training based on a small portion of in-distribution private data. The recipe mainly consists of training a diffusion model with data augmentation, and optimizing towards flat minima through SAM and weight multiplicity. Experiments on CIFAR and ablation study verified the effectiveness of the recipe.

**Strengths:**

- Algorithmic advancement in both parts of the two-stage private training framework (warm-up on public dataset + continued private training)
- The ablation study (table 1) is clearly very useful

**Weaknesses:**

- The presentation could be improved: consider adding a figure to illustrate the two-stage private learning framework, so that it would be easier to understand the contribution of your work and facilitate future algorithmic development.
- I didn't fully understand the interpretation and implication of Theorem 4.1. It claims that "when $r$ is small, $p_\theta$ that minimizes $D_{KL}(p_{pub} || p_\theta)$" can approximate $p_{data}$ and $p_{pr}$" --- I don't see why this is the case; as a matter of fact, when $r$ is extremely small, the signal will be so weak that it is impossible for $p_\theta$ to learn the actual data distribution. The authors also mentioned "a more accurate approximation within the in-distribution data can enhance the model's ability to match the whole data distribution". I think this is only true if the in-distribution data are sufficiently diverse; as demonstrated in Section 4.1, fitting a generative model on limited data will easily lead to overfitting.
- The intuition behind weight multiplicity is not well-explained and the connection with augmentation and noise multiplicity is not sufficiently discussed. I don't see why it can improve generalization; as a matter of fact, from Table 1 it seems that the marginal gain of introducing weight multiplicity is very small (0.5%). As a side note, such a minor improvement is actually the mixed effect of weight multiplicity + extended training (i.e., using the public gradient in private training). The authors should fix this issue and isolate out the effect of weight multiplicity.
- The experiments should be running over multiple random seeds, and please attach the standard deviations along with the mean values.
- The writing can be improved. There are multiple grammatical errors and strange phrasing, for instance, "well-generalizing" -> "well-generalized"; "Explicit diversity with data augment" -> "data augmentation"; "we need to larger the weight" -> "we need to increase the weight"; "flat minima to public data" -> "flat minima of the loss of public data"; "suggest weight multiplicity" -> "propose weight multiplicity". Make sure to do a professional proofreading before submitting your work.

**Questions:**

See above